# Increased Expression of Circulating Stress Markers, Inflammatory Cytokines and Decreased Antioxidant Level in Diabetic Nephropathy

**DOI:** 10.3390/medicina58111604

**Published:** 2022-11-07

**Authors:** Ghazal Mansoor, Muhammad Tahir, Tahir Maqbool, Sana Qanber Abbasi, Faheem Hadi, Tania Ahmad Shakoori, Shabana Akhtar, Muhammad Rafiq, Muhammad Ashraf, Inam Ullah

**Affiliations:** 1Centre for Research in Molecular Medicine, Institute of Molecular Biology and Biotechnology, The University of Lahore, Lahore 54660, Pakistan; 2Department of Physiology, Shareef Medical and Dental College, Lahore 54000, Pakistan

**Keywords:** type 2 diabetes, diabetic nephropathy, GSH, IL6, AOPPS, MDA, MPO, AGEs

## Abstract

*Background and Objectives:* The main objective of the present study was to determine the role of oxidative markers (glutathione (GSH), advanced oxidation protein products (AOPP), advanced glycation end products (AGEs), and malondialdehyde (MDA)) and inflammatory biomarkers (interleukin-6 IL-6, tumor necrosis factor α (TNF-α), myeloperoxide (MPO)) in the development of diabetic nephropathy along with routinely used biochemical parameters. *Materials and Method:* This was a case control study. All the selected patients were screened and enrolled by convenient non-probability sampling technique at the Jinnah hospital in Lahore. Informed consent was obtained before enrollment of the study subjects. A total of 450 patients enrolled in the study, and they were divided into three groups, 150 subjects with type 2 diabetes and 150 diagnosed diabetic nephropathy (DN) vs. 150 healthy individuals as a control group. Five mL of venous blood sample was taken from the antecubital vein of each participant. Statistical analysis was performed by SPSS. The results of all variables were evaluated by using one way ANOVA. *Results:* The mean value of biochemical parameters (WBCs, platelets, prothrombin time, HbA1c, glucose, urinary albumin-to creatinine ratio (UACR), triglycerides, LDL, HDL, serum creatinine, urinary albumin (creatinine)) were increased and Hb (g/dL), red blood cells (RBCs), hematocrit (Hct), free serum insulin levels, and estimated glomerular filtration rate (eGFR) were decreased in the nephropathy group compared to the control and type 2 diabetes groups. The mean values of MDA, AGE, and AOPPs in type 2 diabetes and diabetic nephropathy were significantly increased compared to the control group. GSH level was decreased in type 2 diabetics and DN patients as compared to the control group. In addition, IL-6, TNFα, and MPO levels were also increased in case of diabetes nephropathy compared to controls. *Conclusions:* ROS mediated injuries can be prevented by the restoration of an antioxidant defense system, through the administration of antioxidant agents. Moreover, increased levels of inflammatory mediators are responsible for enhancing inflammation in patients with diabetic nephropathy.

## 1. Introduction

Diabetes mellitus (DM) is a metabolic syndrome with abnormality in the metabolism of carbohydrates, protein and lipids, and which is characterized by an absolute and relative deficiency of insulin secretion [1]. Through recent studies, it has become clear that diabetes is a primary factor in morbidity and mortality worldwide [2]. Cardiomyopathy, nephropathy, retinopathy, hepatopathy and neuropathy are the most common complications. Diabetes is classified into two main types: type 1 Diabetes mellitus is characterized by the complete absence of insulin caused by the destruction of beta cells of Islets of Langerhans of the pancreas due to autoimmune defect while type 2 diabetes mellitus is characterized by compensated weakness of beta cells due to receptor defect or the poor quality of insulin due to toxic agents or viral infection [3].

One of the most common and serious complications of DM is diabetic nephropathy (DN) with an alarming increase in its prevalence particularly in developed countries. It is characterized by morphological and functional changes in the body. These changes ultimately lead to high-density lipoprotein (HDL), cholesterol, and lipoprotein levels of triglycerides being disturbed. High levels of total and non-HDL cholesterol, as well as low levels of HDL cholesterol, have been significantly associated with an increased risk of kidney dysfunction. Insulin deficiency may influence the level of free fatty acid. There is increasing evidence that an ongoing cytokine-induced inflammatory response is related closely to the pathogenesis of diabetic nephropathy. It is also associated with elevated levels of urinary albumin, creatinine, and urinary-to-creatinine ratio. One of the most accepted hypotheses describing the complications of diabetes is the role of oxidative stress and inflammatory factors. It is hypothesized that an increase in glucose level enhances oxidative stress which markedly changes the structure and function of lipids and proteins by inducing peroxidation and glycoxidation [4,5]. Therefore, increased levels of glucose cause auto-oxidation and glycation of proteins and stimulate the polyol pathway [6]. Furthermore, increased levels of glucose stimulate auto-oxidation and glycosylation of proteins leading to the formation of free radicals which increase the reactive oxygen species, ultimately leading to a decrease in the activity of antioxidants—a key factor in oxidative stress [7].

Macrovascular and microvascular issues are caused by alterations in functions of beta cells through various signaling pathways as shown in Figure 1 as well as the formation of ROS which leads to a decline in antioxidant activity. An activated nuclear transcription factor, nuclear factor kappa-light-chain-enhancer of activated B cells (NF-𝜅B) defects are the main streamline factors that lead to complications associated with diabetes. These factors do so by promoting stress related genes by the involvement of several ROS associated signaling pathways which ultimately end up in transcription of pro-inflammatory proteins. Activation of several mechanisms such as interleukins (IL-1β and 6), macrophage chemotactic proteins (MCP-1), tumor necrosis factor (TNF-𝛼), and pro-inflammatory chemokines are involved in the further progression of diabetic complications. Through the Bax–caspase pathway, activation of signaling pathways is caused by an increased ROS level secondary to an increase in glucose levels that lead to decrease in electrochemical gradient by the leakage of mitochondrial cytochrome into the cytoplasm which activates apoptosis [3]. The role of inflammation and inflammatory cytokines in causing diabetic complications, especially DN, has recently gained popularity. A marked increase in the activity of T cells and an abnormal expression of T cell cytokines have been implicated in DN [8,9].

The purpose of the current study was to evaluate the role of oxidative markers AOPP, AGEs, MDA antioxidant GSH and inflammatory biomarkers IL-6, TNF-α, and MPO in diabetic nephropathy compared to controls and type 2 diabetics. According to the findings all the stress and inflammatory markers were elevated and antioxidant levels were decreased in diabetic nephropathy groups and type 2 diabetics compared to control, which could indicate the risk factor for progression of the disease.

## 2. Materials and Methods

### 2.1. Study Aims

The general goals of the study of diabetic nephropathy are to avoid acute decompensation, prevent or delay the severe loss of function of the kidney, decrease mortality, and maintain a good quality of life in both males and females of diabetic nephropathy.

### 2.2. Study Participants

All the patients (450) were screened at the Jinnah hospital in Lahore. Informed consent was obtained before being included in this study. A total of 450 patients were enrolled in the study divided into a diagnosed type 2 diabetic group of 150 patients (D), a diagnosed diabetic nephropathy group (DN) of 150 patients and 150 healthy individuals as a control group.

### 2.3. Sampling Technique

Convenient non-probability sampling technique was employed to select the study participants.

### 2.4. Inclusion and Exclusion Criteria

Participants who accepted to take part in the study were recruited. Selected subjects were between 30–50 years. They were further classified into three groups with 150 patients in each group: a control group, a type 2 diabetics group, and a diabetic nephropathy group. All diabetic and healthy people affected with other diseases or under medication that can affect oxidative stress markers were excluded from the study. Diabetic patients were on a low-carbohydrate diet and treated with insulin. The control group was made up of healthy volunteers present either in the institute or in family. Moreover, none of the control individuals had a history of chronic infections or metabolic dysfunction such as hypertension, diabetes, and cancer.

None of the control subjects were taking any medication.

### 2.5. Determination of Malondialdehyde (MDA)

The method of Okawa et al. (1979) was used to determine the levels of MDA spectrophotometrically. Sodium Dodecyl Sulfate (SDS) (8.1%), Thiobarbituric acid (TBA), acetic acid (20%) with pH 3.5, n-Butanol, TBA (80%), and distilled water were used as reagents. During this protocol, a 200 µL serum sample was taken into the test tube and added SDS with a concentration of 8.1%. Then, 1.5 mL of acetic acid and 1.5 mL of TBA solution were added into the mixture. After that, distilled water was added and a 4.0 mL mixture was prepared. Then, the prepared mixture was heated in a water bath for 60 min at 90 °C, then chilled with tap water and 5.0 mL of n-butanol and 1.0 mL of distilled water were added. The mixture was shaken vigorously and centrifuged for 10 min at 4000 rpm. The upper layer was collected and an absorbance was taken in the spectrophotometer at 532 nm.

### 2.6. Determination of Glutathione (GSH)

The levels of GSH were measured by the method of Moron et al. (1979). Commonly, GSH joins with nitrobenzoic acid and oxidized glutathione which consequently synthesizes chromophore TNB at the absorbance of 412 nm. This protocol required reagents including TCA (5%), DTNB (0.2 M), standard GSH (5%) and phosphate buffer (0.2 M). An amount of 0.1 mL of supernatant was prepared up to 0.2 M sodium phosphate buffer at 8.0 pH. Furthermore, the standard GSH was prepared for 2–10 moles. In addition, 2 mL of DTNB solution was added to the mixture and a yellow color then appeared. The absorbance was taken through a spectrophotometer at 412 nm and GSH was expressed via nmol for every sample size.

### 2.7. Evaluation of Advanced Oxidative Protein Products (AOPPS)

The levels of AOPPs were determined by the protocol of Witkosarsat et al. (1996). In this procedure, the serum sample was analyzed by semi-automated method. Moreover, the levels were measured on microplate reader through spectrophotometer. After this step, it was calibrated through chloramine T solution and then potassium iodide was added to take the absorbance to 340 nm. Plasma diluted PSB was added in the concentration of 200 mL in 96-well microtiter plates. The levels of AOPPs were expressed by micromoles per liter of chloramines.

### 2.8. Determination of Advanced Glycation Endproducts (AGES)

In vitro, AGE-HAS was performed according to [10,11], AGE-HSA was made by incubating HSA (type V; Sigma, St. Louis, MO, USA; 50 mg/mL) with 500 mM glucose in PBS for 65 days at 37 °C. TCA precipitated plasma proteins or AGE-HSA. It was then dissolved in 250 mL 0.01 M heptafluorobutyric acid (Sigma). Then, 4 mg plasma protein was injected into an HPLC apparatus (Waters Division of Millipore, Marlborough, MA, USA), 30.46 cm C18 Vydac type 218 TP (10 mm) (Separations Group, Hesperia, CA, USA). From 0 to 35 min, HPLC was designed with a 10% acetonitrile gradient. Pentosidine was eluted in approximately 30 min using 335 nm excitation and 385 nm emission fluorescence.

### 2.9. Determination of IL-6 and TNF-α by ELISA Kit Method

The levels of IL-6 and TNF-α were determined by the human available diagnostic ELISA kit method. The standard was prepared from 200 pg/mL and assessable concentration of interleukins and TNF-α remained at 3 pg/mL. First of all, 100 µL of serum sample was added to the ELISA plate and incubated at room temperature for 120 min. After incubation, the plate was washed with washing buffer solution. After the removal of extra water from the ELISA plate, the plate was inverted on a paper towel. An amount of 100 µL of HRP conjugate solution was added into each well and incubated at room temperature for 1 h. The plate was washed again and dried on a paper towel for the removal of residual water. After that, the substrate was added into each well with a concentration of 100 µL and kept in dark room temperature for incubation for a period of 15 min. Later on, TMB was added with the amount of 100 µL into each well and placed for one hour. Finally, 50 µL of stop solution was added which provided the color perception during this reaction and showed the presence of TNF-α and interleukins in the serum sample of patients with diabetic nephropathy. Finally, the absorbance was taken at the 460 nm wavelength by ELISA reader.

### 2.10. Determination of MPO by Using ELISA Kit Assay

Human Elisa Kit [ABCAM] was used to measure the levels of MPO. All the materials and reagents were prepared at room temperature. An amount of 50 µL of serum sample, standard and blank were put into their respective wells. After that, 50 µL of antibody mixture was added into each well and incubated at room temperature for 1 h. For the removal of the mixture, the microplate was washed with washing buffer and by tapping the plate on a paper towel for the removal of residual fluid from the plate. After washing, 100 µL of substrate was added into each well and the plate was placed for incubation at 37 °C. Again, the plate was washed with washing buffer and this process was repeated three times. After washing, 100 µL of stop solution was added into each well which generated a yellow shade color. Then absorbance was taken on ELISA plate reader at 450 nm.

### 2.11. Estimation of Insulin and Glucagon

Insulin was measured with the help of double anti-body immunoprecipitation technique explained by Hales and Randle (1963), while glucagon was estimated by radioimmunoassay method of Sakurai and Imura (1973).

### 2.12. Meaurement of Urinary Albumin and Urinary Albumin to Creatinine

The concentration of urinary albumin and urinary albumin to creatinine was determined using a Sequoia-Turner Digital Fluorometer, Model 450. For this analysis, the urine sample was collected from the subjects according to the study protocol. There are no particular instructions, such as special diet or fasting, that are required. The optimum specimen tube was selected (3–5 mL screw top cryogenic vial) for urine sample storage. All the materials and reagents were prepared at room temperature. Distilled water was sterilized, we then filtered the type 1 distilled water through sterile (115 mL), HCL (1 mol/L0), KOH (5 mol/L), KH_2_PO_4_ (0.003 mol/L), phosphate buffer saline stock solution (20× PBS), and buffer saline (IX PBS). Four vials were reconstituted with 5 mL of type 2 water and incubated for 1 h. After incubation, we added 2 mL of type 2 water and dialyzed against 0.003 mol/L KH_2_PO_4_ for four hours. After that, we divided the immunobeads into 6 bottles with a concentration of 250 mL and then incubated overnight. At the final stage, the absorbance was taken at 450 nm.

### 2.13. Measurement of Serum Creatinine

The level of serum creatinine is used to determine the performance of the kidneys and check the blood-filtrating capacity of the kidneys. Creatinine is a chemical compound that exists in the body as a waste product in urine. During analysis, serum, lithium heparin plasma, K2-EDTA TAPS buffer (30 mmol/L), creatinase (332 µkat/L), ascorbate oxidase (33 µkat/L), catalase (1.67 µkat/L) and HTIB (1.2 g/L) were used as reagents. We mixed all of the specimens and allowed them to make a clot after the addition of the serum sample. After that, we centrifuged the combine mixture for 10 min at 2000× *g*. We preincubated the working reagent, standard and sample at room temperature. We adjusted the photometer with distilled water at zero absorbance. We prepared the working reagent with a concentration of 1.0 mL and used the sample/standard with a concentration of 100 µL. At the final stage, we recorded the absorbance at 510 nm with the spectrophotometer.

### 2.14. Measurement of Low Density Lipoprotein (LDL)

LDL cholesterol is catalyzed into fatty acid and free cholesterol through cholesterol esterase. Cholesterol oxidase oxidized the cholesterol into cholestone and hydrogen peroxide. This hydrogen peroxide merges with 4-aminophenazone and phenol in the presence of peroxidase. As a result, a purple colored is generated and the color intensity expresses the concentration of cholesterol.

### 2.15. Determination of High Density Lipoprotein (HDL)

The concentration of HDL can be measured with the help of HDL cholesterol assay. Magnesium chloride (25 mM/L) and phosphotungstic acid (0.55 mM/L) were used as reagents. All of these reagents were added into a test tube with their respective concentrations and shaken vigorously. The mixture was then incubated at room temperature for 10 min. The supernatant was removed after centrifugation. The color intensity expressed the concentration of cholesterol liquid reagent.

### 2.16. Estimation of Triglycerides (TG)

Three test tubes were taken and named as blank, calibrator and assay tube. The buffer solution with concentration of 300 µL was added into watch test tube. Enzymes (lipase (≥1000 IU/L), glycerol 3 phosphate oxidase (≥3000 IU/L), POD (≥1700 IU/L) and glycerol kinase (≥600 IU/L)), buffer (magnesium chloride (9.8 mmol/L), chloro-4-phenol (3.5 mmol/L) and PIPES (100 mmol/L), 4-amino-antipyrine (PAP) 0.5 mmol/L, standard (triglycerides (200 mg/dL) and glycerol (2.8 mmol/L)), and adenosine triphosphate Na (1.3 mmol/L) were used as reagents during this assay. These contents were shaken and incubated at room temperature (10 min). The absorbance was taken with a spectrophotometer (546 nm). Triglycerides levels were expressed by unit mg/dL.

### 2.17. Evaluation of CBC (Complete Blood Count)

Complete blood count of the selected subjects was performed using an automated hematology blood analyzer by Sysmex (version. XP-2100).

### 2.18. Statistical Analysis

All data of experimental groups were expressed as mean ± SEM. For statistical analysis, group means were compared by one-way ANOVA and Bonferroni’s test was used to identify differences between groups by using graph pad prism. A *p*-value less than 0.05 was considered significant from statistical analysis. Endnote was used to insert references.

## 3. Results

The current study was designed to investigate the role of inflammatory cytokines, oxidative stress and cellular stress response in patients suffering from type 2 diabetes and in age matched healthy subjects. The total number of individuals recruited into the study was 450, including 150 patients with type 2 diabetes, 150 diabetic nephropathy patients, and 150 normal healthy controls. All the participants in the current study were matched for age, sex and body mass Index (BMI). Duration of type 2 diabetes and fasting blood sugar levels were similar in all the groups. The demographic profile of the patients and controls is summarized in Table 1 and Figure 2.

Further biochemical parameters were performed, where decreased levels of Hb, RBCs, HCT, free serum insulin and eGFR were observed and increased levels of WBCs, platelets, prothrombin time, HbA1c, glucose, Urinary albumin-to creatinine ratio, triglycerides, LDL, HDL, urinary albumin creatinine, and serum creatinine were observed. Biochemical parameters involved in current study are shown in Table 2 and Figure 3, Figure 4 and Figure 5.

An increased level of all the stress markers and inflammatory markers while the decreased level of antioxidant GSH was observed in diabetic nephropathy which was further increased in the diabetes type II group. The mean values of MDA in diabetics and DN patients were 6.30 ± 2.32, and 7.30 ± 2.60 nmol/mL, respectively while in the control group mean MDA level was 3.16 ± 1.21 nmol/mL which was found to be statistically significant. The levels of GSH in diabetics and DN patients were decreased as compared to the control group. The data interpretation of AOPPs also showed significantly increased levels in diabetics, DN group 24.52 ± 8.47 mmol/L, 28.25 ± 7.73 mmol/L as compared with normal subjects 15.90 ± 5.3 mmol/L. The mean serum level of AGEs in type 2 diabetes, DN disease patients and normal people was documented as 2.78 ± 0.97 U/mL, 11.46 ± 0.98 U/mL, and 1.33 ± 0.57, respectively, presenting an increased level in type 2 diabetes and DN patients in comparison to control subjects. The levels of IL-6 were significantly increased in the patients with type 2 diabetes, and diabetic nephropathy as compared to control individuals. The data analysis of TNF-α and MPO showed statistically significant enhanced levels in the diabetic and DN group comparison with normal as shown in the Table 3 and Figure 6.

## 4. Discussion

Diabetic nephropathy is a condition characterized by the uncontrolled secretion of urine albumin, loss of glomerular filtration rate, and glomerular lesions. Different epidemiological studies have demonstrated that family history ethnicity, abnormal hematological profile, gestational diabetes, elevated blood pressure, dyslipidemia, and obesity are the major risk factors for diabetic nephropathy [10]. Other putative risk factors include external environmental pressures such as smoking and inhalation of other toxins, elevated glycosylated hemoglobin levels [HbA1c], proteinuria and elevated systolic pressure [11,12]. Although nephropathy in diabetic conditions is the strongest predictor of mortality in patients, deregulation of the local metabolic environment triggered by oxidative stress and inflammation, and subsequent remodeling of tissue, are the main causes of kidney failure [13].

The prevalence of type 2 diabetes is increasing at an alarmingly fast pace. One of the leading causes of diabetes is an end-stage renal disease such as diabetic nephropathy (post type 2 diabetes). Oxidative stress triggered by hyperglycemia plays an important role in the pathogenesis of DN.

In the present study, we performed analysis of different stress markers, inflammatory cytokines, and antioxidants along with biochemical parameters and demographic data. We found elevated MDA levels in the diabetic nephropathy group compared to control and type 2 diabetes. As mentioned in a previous study, MDA is formed by lipid peroxidation and causes changes in macromolecule up regulation of MDA that could be the cause of multiple diseases [14,15]. A study by Verma et al. in 2014 found increased production of MDA in Diabetics and DN patients in comparison to the controls [16]. Rani et al. also reported in 2019 that diabetic patients suffer more from oxidative stress and the stress is even higher in DN patients than control. The study reported a significantly higher level of MDA in complicated type 2 diabetes with nephropathy and non-complicated type 2 diabetes as compared to the healthy controls [17]. A study by Hou et al. in 2021 observed increased levels of MDA type 2 diabetic retinopathy patients. From all three groups, type 2 diabetes with complication, DM without complication, and the control healthy group MDA level was significantly higher in DN [18]. Data of previous studies correlate with the current study.

In the current study, GSH levels were found to be highest in the control group, as compared to the diabetics and DN patients. A study by Calabrese et al. (2012) pointed out that type 2 diabetic patients are under severe systemic oxidative stress. They compared the content of reduced and oxidized GSH in the plasma of type 2 diabetic patients to check the antioxidant status. The study concluded that there was a 63% decrease in reduced GSH levels as compared to the controls, whereas a rise of 46% was observed in the content of oxidized GSH in diabetic samples as compared to the control samples [19]. Miranda-Díaz et al. showed that in DN patients there is an imbalance in prooxidant/antioxidant processes. The ROS diminishes the enzymatic activity of glutathione peroxidase which markedly decreases GSH levels in DN [20]. The results of the abovementioned studies are comparable with our study results where a decreased GSH level was observed in the DN group.

Results of the current study are in accordance with previous studies. According to Conti et al. (2019) evaluated levels of AOPPs in a cohort of diabetes patients and hypertensive patients was observed. The study concluded that oxidative stress was highest in diabetics as compared to the healthy controls. Also, higher levels of AOPPs were found in DM patients in comparison to the controls, and the index was even significantly higher in patients with diabetic nephropathy than in DM patients without nephropathy [21]. In the current study a significantly higher level of AOPPs in DN group was found compared to controls. According to another study by Kar et al. (2014) AOPP in 50 diagnosed diabetic patients was high compared to 47 age- and sex-matched controls [22]. In another study by Sharada et al. (2012) the levels of AOPP in the plasma of type 2 diabetes were not only raised in type 2 diabetes patients but also increased progressively with the development of DN [23,24]. All these studies strengthen the findings of the present study.

Several clinical studies have shown the same results as our study and estimated that the levels of advanced glycation end products (AGEs) were significantly highest in DN patients compared to the control group. A study by Nishad in 2021 analyzed the association of serum AGEs with impaired kidney function in type 2 diabetes. They found a positive correlation between AGEs and impaired kidney function in type 2 diabetes patients and suggested that AGEs can serve as prognostic markers for DN [25]. Another study in 2022 indicated that higher levels of AGEs predicted poor morphological features in DN with type 2 diabetes. In tissues and serum of type 2 diabetes, AGE levels were considerably higher than in healthy controls [26]. Further, the levels of AGE were double in DM patients with renal disease as compared to DM patients without renal illness [27]. Yamagishi et al. (2010) reviewed the role of AGEs and their receptors in DN. The results revealed that both the serum and tissue levels of AGEs were significantly raised in type 2 diabetics in comparison to non-diabetic controls. Also, patients had twice the concentration of AGEs. They also pointed out that the degradation products of AGEs increase in diabetics and non-diabetic subjects with renal disease [28]. All of these studies are comparable to the findings of our study.

We found that levels of IL-6 were significantly higher in the DN group as compared to the controls and with type 2 diabetes, also we observed that the levels of IL-6 in DN group were double the levels found in the diabetic group. Kreiner et al. (2022) investigated the role of IL-6 in diabetes, CKD and CVD. They found out that IL-6 exerts proinflammatory effects in the pathophysiology of both diabetes and CKD. It has been suggested that IL-6 induces systemic chronic inflammation via its proinflammatory effects, which is central in the pathophysiology of type 2 diabetes [29]. A cross sectional study was conducted by Ha et al. (2021) which aimed to find out the correlation of IL-6 levels between type 2 diabetes and diabetic nephropathy. The study included 59 type 2 diabetes patients with 30 patients of DN and 29 non-DN patients. Their results showed that the levels of IL-6 were significantly higher in type 2 diabetes with diabetic nephropathy as compared to the non-DN group [30]. A study by Sindhughosa et al. also pointed out the role of IL-6 in type 2 diabetes and DN. They concluded that serum IL-6 levels increased in type 2 diabetics with nephropathy in comparison to the diabetics without nephropathy [31,32]. The abovementioned studies stand in contrast to our study results as, in our study, the highest levels of IL-6 were observed in the DN group.

In the present study, TNF-α (pg/mL) was also estimated in three groups. The levels were highest in the DN group followed by the diabetic group as compared to the healthy controls. According to previous studies, the serum concentrations of TNF-α levels were increased in the type 2 diabetes group but were highest in the type 2 DN group [33,34]. A study by Yeo et al. and El-Badawy et al. suggested that TNF-α may serve as an independent risk factor for CKD in patients with type 2 diabetes [35,36]. All these studies correlate with the results of our study.

The present study estimated the levels of Myeloperoxidase in the study groups. It was observed that levels were raised in the DN group as compared to the controls and the type 2 diabetes group. In a study by Moneam et al. (2021) serum MPO levels were significantly raised in type 2 diabetes as compared to the controls and a positive correlation was found between MPO levels, glycated hemoglobin (HbA1C) and FBG [37]. Rovira-Llopis et al. (2013) investigated the role of MPO as a key component in ROS-induced vascular damage related to nephropathy and type 2 diabetes. They concluded that serum MPO levels were high in type 2 diabetes and patients with nephropathy presented even higher MPO levels [38,39]. The results of the abovementioned studies correlate with the current study results.

We also performed biochemical parameters which are routinely performed in labs to check diabetes and diabetes nephropathy and observed decreased levels of Hb, RBCs, HCT, free serum insulin and eGFR, as well as increased levels of WBCs, platelets, prothrombin time, HbA1c, glucose, urinary albumin-to-creatinine ratio, triglycerides, LDL, HDL, urinary albumin creatinine, and serum creatinine.

Treatment of diabetic kidney disease will surely have a better outcome if the modalities used for treatment are well targeted. Although good glycemic control may be the best prevention of DN, it can develop in spite of the treatment of diabetes. Such targeted modalities can be deduced by the knowledge of powerful antioxidants. Inhibitors of oxidative stress and inflammation should provide useful targets for therapy. Therefore, this article was focused on the basic mechanisms of the oxidative stress induced by diabetes mellitus through ROS formation as well as various signaling pathways that are responsible for the activation of different downstream signaling cascades that ultimately lead to functional and structural changes in the kidney. ROS-mediated injuries can be prevented by the restoration of the antioxidant defense system which can be achieved by the administration of antioxidant agents. Moreover, inhibition of the inflammatory mediators can be achieved to reduce or prevent diabetic nephropathy.

### 4.1. Study Limitations

A limitation of this study is the cross-sectional design, which only provides the basis for associations and does not evaluate the ‘cause and effect’ relationship between elevated circulatory stress markers and inflammatory markers.

### 4.2. Strength of the Study

Diabetic nephropathy remains a major challenge in the field of medicine. In this study we summarized the critical signaling pathways and biological processes involved in DN. A large amount of evidence now exists to prove that several proinflammatory cytokines are known to be involved in its mechanism, including MPO, IL-6, and TNF-α.

### 4.3. Future Recommendations

It is plausible to hypothesize that novel therapeutic approaches can be designed to enhance circulatory stress response as a mean to control and reduce oxidative stress-mediated formation of ROS as well as to remove ROS-induced modifications, thus constituting an important component of future prophylaxis and therapy in patients with diabetes.

## Figures and Tables

**Figure 1 medicina-58-01604-f001:**
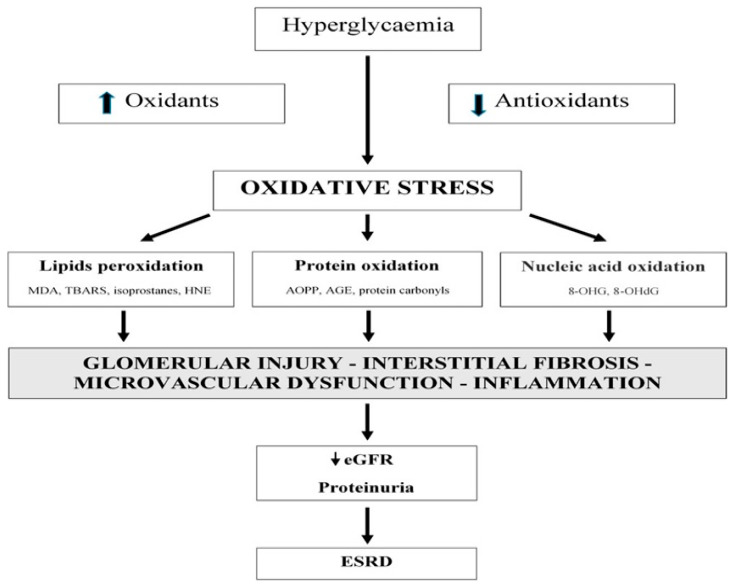
Increased glucose level can lead to end-stage renal disease. Hyperglycemia leads to oxidative stress which in turn activates elevated levels of lipid peroxidation, protein oxidation, and nucleic acid peroxidation that leads toward glomerular injury, then interstitial fibrosis to microvascular dysfunction and finally inflammation which further downregulate eGFR proteinuria and finally end-stage renal disease.

**Figure 2 medicina-58-01604-f002:**
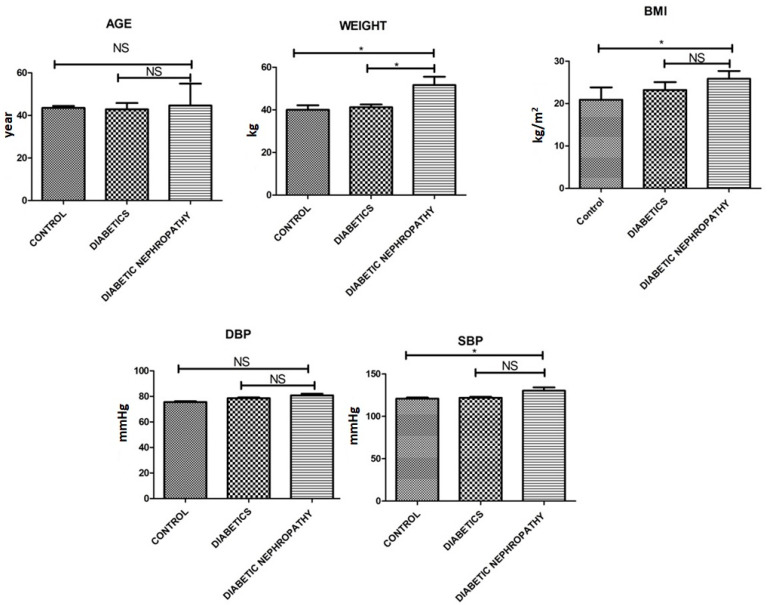
Age, weight, BMI, systolic and diastolic blood pressure of all the individuals. There is a significant difference between control and diseased group, an elevated level of BMI and weight can be observed between control vs. diseased group. Where *p* ≤ 0.05, * (less significance) represents the significant difference between groups.

**Figure 3 medicina-58-01604-f003:**
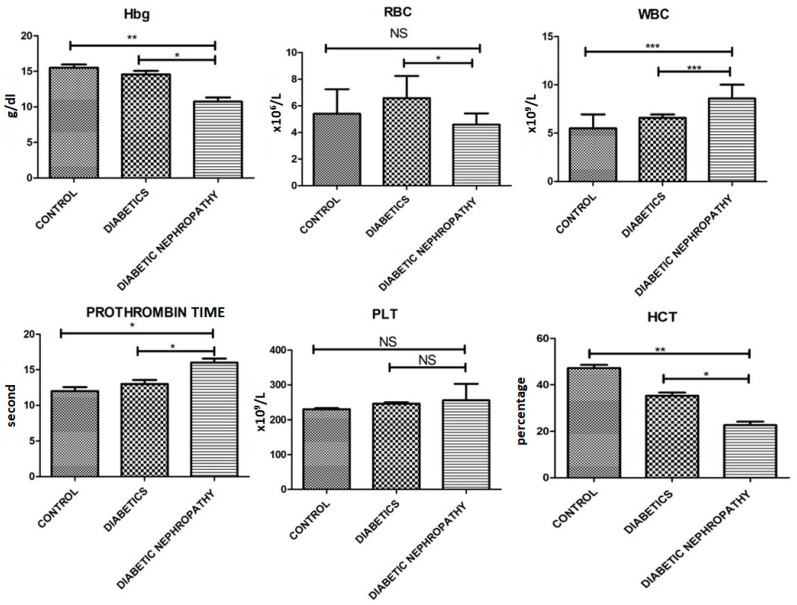
Hbg, RBCs, WBCs prothrombin, platelets, HCT. Of all the individuals there is a significant difference between control and diseased group between all the groups except platelets. Where *p* ≤ 0.05. *, **, *** (less, moderate, highly significant) represents the significant difference between groups.

**Figure 4 medicina-58-01604-f004:**
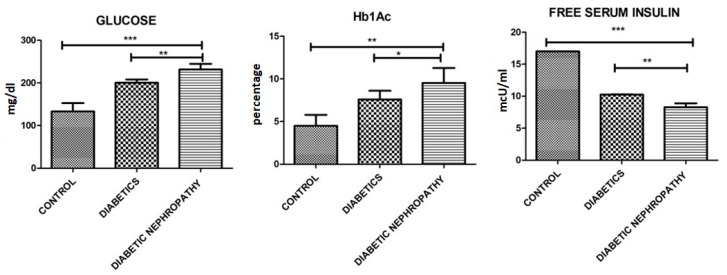
Glucose, Hb1Ac, and free serum insulin. Of all the individuals there is a significant difference between control and diseased group and elevated levels of all the parameters can be observed in DN compared to control and to type 2 diabetes groups. Where *p* ≤ 0.05. *, **, *** (less, moderate, highly significant) represents the significant difference between groups.

**Figure 5 medicina-58-01604-f005:**
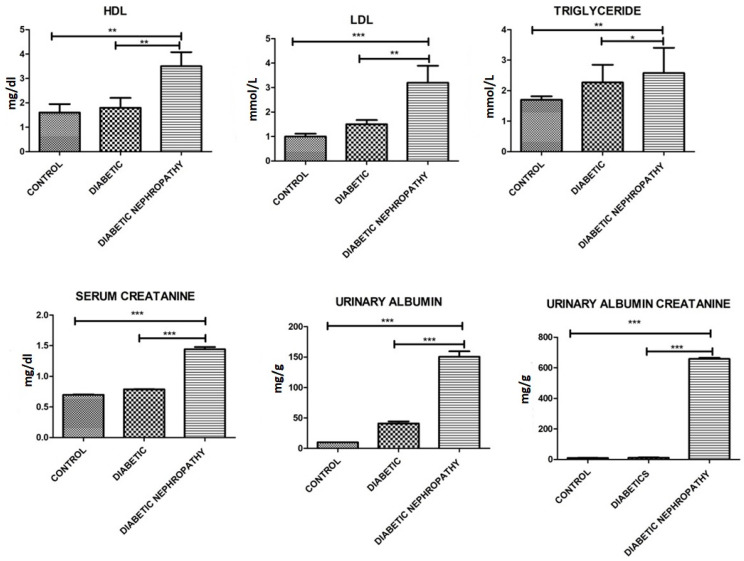
Graphical presentation of HDL, LDL, triglyceride, serum creatinine, urinary albumin, and urinary albumin creatinine. Of all the individuals there is a significant difference between control and diseased group. Elevated levels of all the parameters can be observed in DN compared to control and to type 2 diabetes groups. Where *p* ≤ 0.05. *, **, *** (less, moderate, highly significant) represents the significant difference between groups.

**Figure 6 medicina-58-01604-f006:**
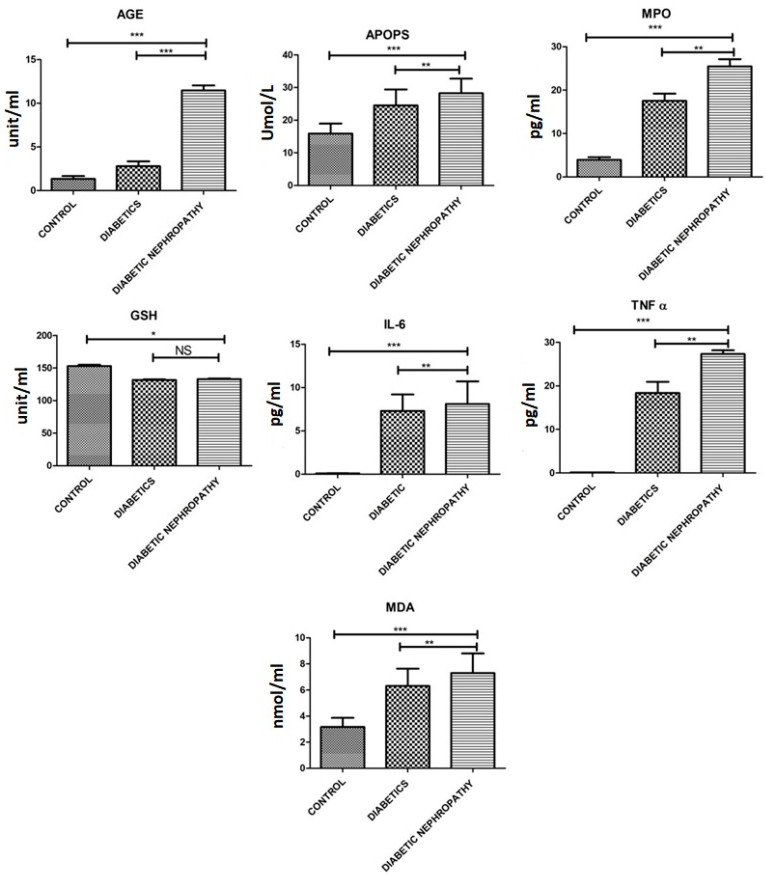
Levels of AOPPS, AGEs, TNFa, GSH, MPO, MDA and IL6 of all the individuals. There is a significant difference between control and diseased groups. Elevated levels of all the stress markers and decreased levels of GSH can be observed in the diseased group compared to control while the stress markers are highly increased in the diabetic nephropathy group. Where *p* ≤ 0.05. *, **, *** (less, moderate, highly significant) represents the significant difference between groups.

**Table 1 medicina-58-01604-t001:** Demographic profile of diabetics, diabetic nephropathy. Subject vs. healthy age matched control.

Variable	Control (*n* = 150)	Diabetics (*n* = 150)	Diabetic Nephropathy (*n* = 150)	*p* Value
Weight [kg]	40.25 ± 3.3	41.22 ± 2.25	51.61 ± 6.8	0.001
Age [Yrs.]	43.55 ± 1.5	42.85 ± 5.15	44.61 ± 7.89	0.054
BMI [kg/m]^2^	20.88 ± 5.1	23.22 ± 3.2	25.84 ± 3.2	0.03
Systolic BP [mmHg]	120.99 ± 2.2	121.91 ± 2.24	130.33 ± 6.75	0.51
Diastolic BP [mmHg]	75.55 ± 1.15	78.56 ± 1.19	80.75 ± 2.27	0.523

**Table 2 medicina-58-01604-t002:** Biochemical profile of diabetics. Diabetic nephropathy subjects vs. healthy age matched control.

Variable	Control (*n* = 150)	Diabetics (*n* = 150)	Diabetic Nephropathy (*n* = 150)	*p* Value
Hb (g/dL)	15.5 ± 0.78	14.55 ± 0.88	10.76 ± 0.98	0.001
RBCs (×10^6^/uL)	5.4 ± 3.2	6.58 ± 2.88	4.59 ± 1.45	0.05
WBCs (×10^9^/L)	5.5 ± 2.5	6.59 ± 0.59	8.59 ± 2.48	0.0041
PLTs (×10^9^/L)	230.3 ± 5.4	246.3 ± 6.7	255.7 ± 82.0	0.01
Hct (%)	47.2 ± 2.5	35.27 ± 2.45	22.74 ± 2.61	0.0001
Prothrombin time (s)	12 s	13 s	16 s	0.041
HbA1c (%)	4.5 ± 2.55	7.59 ± 1.77	9.55 ± 3.01	0.001
Free Serum Insulin levels (mcU/mL)	17.01 ± 0.02	10.25 ± 0.06	8.29 ± 1.03	0.0014
Glucose (mg/dL)	130 ± 20.0	200.21 ± 13.25	231.33 ± 23.11	0.0001
Urinary albumin-to creatinine ratio (UACR) mg/g	10.22 ± 1.0	11.56 ± 3.58	658.26 ± 12.59	0.0001
eGFR (mL/min)	79.65 ± 6.99	59.45 ± 6.99	41.58 ± 4.78	0.01
Triglycerides (mmol/L)	1.7 ± 0.5	2.27 ± 1.0	3.2 ± 1.0	0.04
LDL (mmol/L)	1.0 ± 0.3	1.5 ± 0.3	3.5 ± 0.8	0.021
HDL (mmol/L)	1.6 ± 0.6	1.8 ± 0.7	3.5 ± 0.5	0.02
Serum creatinine (mg/dL)	0.70 ± 0.01	0.789 ± 0.0024	1.44 ± 0.065	0.001
Urinary Albumin (mg/g Creatinine)	10.12 ± 0.01	41.15 ± 5.56	150.35 ± 15.58	0.0001

**Table 3 medicina-58-01604-t003:** Levels of circulating stress markers, inflammatory cytokines in controls, diabetics, and diabetic nephropathy patients.

Variable	Control (*n* = 150)	Diabetics (*n* = 150)	Diabetic Nephropathy (*n* = 150)	*p*-Value
MDA [nmol/mL]	3.16 ± 1.21	6.30 ± 2.32	7.30 ± 2.60	0.002
GSH U/mL	153.13 ± 3.2	131.79 ± 1.27	133.04 ± 1.68	0.014
AOPPs [mmol/L]	15.90 ± 5.3	24.52 ± 8.47	28.25 ± 7.73	0.0021
AGEs [U/mL]	1.33 ± 0.57	2.78 ± 0.97	11.46 ± 0.98	0.0001
IL-6 [pg/mL]	13.57 ± 2.608	56.73 ± 10.30	110.8 ± 10.30	0.0001
TNF-α [pg/mL]	0.14 ± 0.01	18.39 ± 4.44	27.34 ± 1.45	0.0001
MPO [pg/mL]	3.54 ± 2.2	17.51 ± 2.89	25.47 ± 2.87	0.0001

## Data Availability

All the data is incorporated within the manuscript.

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
