# Peer review of "Increased Expression of Circulating Stress Markers, Inflammatory Cytokines and Decreased Antioxidant Level in Diabetic Nephropathy"

_medicina, 2022, doi:10.3390/medicina58111604_

Round 1

Reviewer 1 Report (New Reviewer)

The paper from Mansoor et al show that Diabetic Nephropathy patients have an increased expression of circulating stress markers, inflammatory cytokines and decreased antioxidant levels. Results are relevant, however the presentation of the results are poor. The authors presents the results as absorbance, that's very odd way to present, so I strongly suggest to all the results being presented with the proper unit of measurement. Moreover I suggest to format the graphs showing the dot blots for better visualization of scatter inside the groups.

Author Response

Title of Manuscript” Increased Expression of Circulating Stress Markers, Inflammatory Cytokines and Decreased Antioxidant Level in Diabetic Nephropathy”

Dear Editors

Thank you for giving us the opportunity to submit a revised draft of our manuscript “Increased Expression of Circulating Stress Markers, Inflammatory Cytokines and Decreased Antioxidant Level in Diabetic Nephropathy to Medicina. We appreciate the time and effort that you and the reviewers have dedicated to providing your valuable feedback on our manuscript. We are grateful to the reviewer for their insightful comments on our paper. We have been able to incorporate changes to reflect most of the suggestions provided by the reviewers. All the changes have been highlighted within the manuscript. Here is a point-by-point response to the reviewer comments and concerns.

Attached below are detailed responses (in blue) to the reviewer comments (in red).  Please let us know if you still have any questions or concerns about the manuscript. We will be happy to address them, now in a timely manner.

Sincerely,

Dr. Tahir Maqbool

Reviewer 1:

Comments & Response

Comment 1: Results are relevant, however the presentation of the results are poor.

Response. We are thankful to the reviewer for fruitful suggestions.  We revised the manuscript and the results are now presented in such a way that the reader may understand it easily

Comment 2: The authors presents the results as absorbance, that's very odd way to present, so I strongly suggest to all the results being presented with the proper unit of measurement. Moreover I suggest to format the graphs showing the dot blots for better visualization of scatter inside the groups.

All the figures with absorbance are now replaced with the figures containing their respective units in the revised version as suggested.

Reviewer 2 Report (New Reviewer)

The Manuscript entitled "Increased expression of circulating stress markers, inflammatory cytokines and decreased antioxidant level in diabetic nephropathy" by Mansoor et al., discusses about the role of oxidative markers like, AGEs and MDA and inflammatory biomarkers; IL-6, TNF-α, MPO in the development of diabetic nephropathy. The patients/subjects were recruited by informed consent of the study subjects. The mean value of biochemical parameters  were increased and RBCs, Hct, Free Serum Insulin levels, eGFR] decreased in  nephropathy group compared to control and diabetes group. The mean values of MDA, AGE, in diabetes and diabetic nephropathy were significantly increased and he level of GSH in diabetics and DN patients were decreased compared to control group. 

The MS looks decent but there are many shortcoming in the present form which needs to rectified.

Major Comments

1. Authors are suggested to improve the english language and grammar errors throughout the MS.

2. Abstract: Clearly state how many total 'subjects were taken in the study. Author stated 300 patients, rather they should mentioned total subjects including the healthy control subjects.

3. Author have stated Hb is decreased at one place and other place they report it is increased in the Abstract, kindly clarify.

4. Authors should mentioned the full form of all the abbreviation at the first mention like, for GSH, MDA and all others.

5. Introduction: line 59-64. 'It is hypothesized that an increase in glucose level enhances oxidative stress which changes the structure and function of lipids and proteins markedly by inducing peroxidation and glycoxidation [provide ref a,b].

Therefore, increased levels of glucose cause auto-oxidation and glycation of proteins and stimulate the polyol pathway [ref c].

ref. a. Antigenic role of the adaptive immune response to d-ribose glycated LDL in diabetes, atherosclerosis and diabetes atherosclerotic patients. Life Sciences, 2016, 151, pp. 139–146.

ref. b.  An immunohistochemical analysis to validate the rationale behind the enhanced immunogenicity of D-ribosylated low density lipo-proteinPLoS ONE, 2014, 9(11), e113144.

ref. c. Advanced Glycation End Products and Diabetes Mellitus: Mechanisms and Perspectives. Biomolecules, 2022, 12, 542.

6. In Materials and methods: line- 154-155: In addition, 4mg of plasma 154 protein was added in HPLC system that is a programmed linear gradient of 17% acetonitrile? where from this sentence came all of a sudden? clarify.

7. AGEs estimation was done in serum or invitro glycated BSA? As per the methodology they have made BSA glycation model with 5000 mM Glucose that too at such high concentration.

8. Result: line 294-295: The mean serum level of AGEs in diabetes, DN disease patients and normal people was documented as 2.78±0.97 U/mL, 11.46±0.98 U/mL, and 1.33±0.57, respectively, presenting an increased level in DN patients in comparison to control subjects.

Which AGEs was estimated here? and where is the methodology for the same?

Secondly, the concentration of AGEs in diabetics is very low, why?

9. Be Uniform in writing Diabetes, Type 2 Diabetes, T-2 DM. There is no uniformity in whole MS.

10. Discussion: line 370-371: . In tissues and serum of type 2 DM, AGEs levels were considerably higher than 370 in healthy controls (citation ref. d.).

ref. d. Acquired immunogenicity of calf thymus DNA and LDL modified by d-ribose: A comparative study.International Journal of Biological Macromolecules, 2015, 72, pp. 1222–1227.

11. Check the spellings throughout the MS. There are many fused words in the MS, kindly sperate them as well to better understand the paper.

Author Response

Title of Manuscript” Increased Expression of Circulating Stress Markers, Inflammatory Cytokines and Decreased Antioxidant Level in Diabetic Nephropathy”

Dear Editors

Thank you for giving us the opportunity to submit a revised draft of our manuscript “Increased Expression of Circulating Stress Markers, Inflammatory Cytokines and Decreased Antioxidant Level in Diabetic Nephropathy to Medicina. We appreciate the time and effort that you and the reviewers have dedicated to providing your valuable feedback on our manuscript. We are grateful to the reviewer for their insightful comments on our paper. We have been able to incorporate changes to reflect most of the suggestions provided by the reviewers. All the changes have been highlighted within the manuscript. Here is a point-by-point response to the reviewer comments and concerns.

Attached below are detailed responses (in blue) to the reviewer comments (in red).  Please let us know if you still have any questions or concerns about the manuscript. We will be happy to address them, now in a timely manner.

Sincerely,

Dr. Tahir Maqbool

Reviewer 2.

Comment 1. Authors are suggested to improve the english language and grammar errors throughout the MS.

Response. The revised manuscript is reviewed by an expert English editor as suggested by the reviewer and has now been much improved.

Comment 2. Abstract: Clearly state how many total 'subjects were taken in the study. Author stated 300 patients, rather they should mentioned total subjects including the healthy control subjects.

Response. Thank you so much for the identification of this important point. 300 is now replaced with the correct number “450” and has been incorporated accordingly.

Comment 3. Author have stated Hb is decreased at one place and other place they report it is increased in the Abstract, kindly clarify.

Response. All the authors would like to appreciate the reviewer for his keen observation while reading this manuscript. Hb was decreased in diabetes and diabetic nephropathy group as compared to control. As mentioned in the result. In abstract it’s mentioned by mistake which is now corrected.

Comment 4. Authors should mentioned the full form of all the abbreviation at the first mention like, for GSH, MDA and all others.

Response. Full forms are now incorporated within the manuscript highlighted in yellow as well track changes as suggested.

Comment 5. Introduction: line 59-64. 'It is hypothesized that an increase in glucose level enhances oxidative stress which changes the structure and function of lipids and proteins markedly by inducing peroxidation and glycoxidation [provide ref a,b].

Therefore, increased levels of glucose cause auto-oxidation and glycation of proteins and stimulate the polyol pathway [ref c].

ref. a. Antigenic role of the adaptive immune response to d-ribose glycated LDL in diabetes, atherosclerosis and diabetes atherosclerotic patients. Life Sciences, 2016, 151, pp. 139–146.

ref. b.  An immunohistochemical analysis to validate the rationale behind the enhanced immunogenicity of D-ribosylated low density lipo-proteinPLoS ONE, 2014, 9(11), e113144.

ref. c. Advanced Glycation End Products and Diabetes Mellitus: Mechanisms and Perspectives. Biomolecules, 2022, 12, 542.

Response. Thanks to the reviewer for suggesting the relevant articles. All the articles  are very relevant to the topic and are now incorporated (cited) within the manuscript at their specific positions and  highlighted.

Comment 6. In Materials and methods: line- 154-155: In addition, 4mg of plasma 154 protein was added in HPLC system that is a programmed linear gradient of 17% acetonitrile? where from this sentence came all of a sudden? clarify.

Response. Dear Reviewer thank you so much for the pointing out this mistake. A proper methodology of AGE-HSA is now incorporated and has been explained within the manuscript. These changes are highlighted and the tracks are changed.

Comment 7. AGEs estimation was done in serum or invitro glycated BSA? As per the methodology they have made BSA glycation model with 5000 mM Glucose that too at such high concentration.

Response. In vitro, AGE-HSA was made by incubating HSA (type V; Sigma; 50 mg/mL) with 500 mM glucose in PBS for 65 days at 37 °C. TCA precipitated plasma proteins or AGE-HSA. It was then dissolved in 250 mL 0.01 M heptafluorobutyric acid (Sigma). Then, 4 mg plasma protein was injected into an HPLC apparatus (Waters Division of Millipore, Marlborough, MA, USA), 30.46 cm C18 Vydac type 218TP (10 mm) (Separations Group, Hesperia, CA, USA). From 0 to 35 min, HPLC was designed with a 10% acetonitrile gradient. Pentosidine was eluted in approximately 30 min using 335 nm excitation and 385 nm emission fluorescence.

Comment 8. Result: line 294-295: The mean serum level of AGEs in diabetes, DN disease patients and normal people was documented as 2.78±0.97 U/mL, 11.46±0.98 U/mL, and 1.33±0.57, respectively, presenting an increased level in DN patients in comparison to control subjects. Which AGEs was estimated here? and where is the methodology for the same?

Response: Glucose-AGE was performed. Methodology of AGE-HSA is incorporated within the manuscript

Comment 9. Secondly, the concentration of AGEs in diabetics is very low, why?

Response. First of all the level of AGEs in diabetes is double to control but less then diabetic nephropathy because During chronic diabetes such as neuropathy, retinopathy, nephropathy, persistent hyperglycemia leads to elevated levels of AGEs in the bloodstream, which induce signaling pathway (Akhtar et al., 2014). Advanced glycation is one of the major pathways involved in the development and progression of different diabetic complications including nephropathy, retinopathy and neuropathy. Tissue and circulating AGE levels are higher in smokers with concurrent increase in inflammatory markers. There is evidence from animal studies that exposure to high levels of exogenous AGEs contributes to renal and vascular complications (Varun Parkash Singh et al., 2014). That’s the reason its multiple times increased in nephropathy compared to control further from diabetes.

Comment 9. Be Uniform in writing Diabetes, Type 2 Diabetes, T-2 DM. There is no uniformity in whole MS.

Response. Dear Reviewer thank you so much for a very much needed point, all the manuscript is thoroughly evaluated. The word diabetes, t-2 DM is replaced with Type 2 diabetes and the uniformity is now maintained all over the manuscript as suggested.

Comment 10. Discussion: line 370-371. In tissues and serum of type 2 DM, AGEs levels were considerably higher than 370 in healthy controls (citation ref. d.).

ref. d. Acquired immunogenicity of calf thymus DNA and LDL modified by d-ribose: A comparative study. International Journal of Biological Macromolecules, 2015, 72, pp. 1222–1227.

Response. Thank you so much for choosing the current and relevant reference. We have incorporated the suggested references accordingly.

Comment 11. Check the spellings throughout the MS. There are many fused words in the MS, kindly sperate them as well to better understand the paper.

Response. These fusions are often occurs due to the differences in the Microsoft word version installed in a machine. We removed the fused words and to overcome on this issue we are sending the revised version in Microsoft word as well as in pdf formats.

This manuscript is a resubmission of an earlier submission. The following is a list of the peer review reports and author responses from that submission.

Round 1

Reviewer 1 Report

Authors have conducted research on the Expression of Circulating Stress Markers, Inflammatory Cytokines and Decreased Antioxidant Levels in Diabetic Nephropathy.

Authors have done this study where they have not much focussed on the biochemical parameters and the type of diabetes.

Authors must define Diabetics, are they type I or type II diabetes? Cannot be expressed in a common term. Because, the authors can do analysis based on the four groups: control, type I, type II and diabetic nephropathy.    

Authors have to focus more on the biochemical parameters such as the basic ones: Fasting blood glucose, postprandial blood glucose level, lipid profiles, Serum creatinine level, Hemoglobin, Albuminuria, Proteinuria etc.

Moreover, the authors need to focus on the drugs taken by the patients

The tables and the figures are represented the same outcomes. Authors can represent the outcomes in the format of tables along with the p values.

The authors have to explore more statistical analysis based on the outcomes.

The authors have to add the study limitations, the strength of the study and future recommendations.

Author Response

Dear Reviewer 1 Thank you for the points to revised manuscript Kindly find the attached file containing answers of all the querries

Introduction part is according to manuscript all the recommended data incorporated except additional assays because current work was my students phD work and she has already publish a paper containing fasting sugar, ldl, hdl, hb1Ac etc reference is included within the manuscript 

rest of all the querries incorporated in the attached word file

Reviewer 2 Report

Increased Expression of Circulating Stress Markers, Inflamma-2 tory Cytokines and Decreased Antioxidant Level in Diabetic 3 Nephropathy

Thank you for the possibility to review this interesting manuscript. The paper concerns an important issue of inflammatory and oxidative stress markers in diabetic patients. MDA, AOPPS, TNFalfa, Il-6 levels were significantly higher in diabetic subject compared to non-diabetic ones. Moreover, there was a difference between diabetic patients with and without nephropathy. The study is interesting, and well designed, but the paper needs some corrections.

1.     The study presents differences in three subgroups however statistical significance (including p value) is not clearly described. Please add the data.

2.     Moreover, one cannot conclude on prognostic value after presenting changes in the concentration between subjects. I suggest adding the analysis showing AUC and the cut-off values of markers predicting higher risk of nephropathy in comparison to diabetic non-nephropatic patients. 

3.     In the Figure 3 one to three stars are used to show the significance. It would be better to present p in number – clearer for the readers. Please change it

4.     The markers used in the manuscript are highly sophisticated and it increases the importance of the paper in current literature. However, these markers are difficult in daily evaluation of patients. The possibility to use less sophisticated methods of analysis may be beneficial to include in the Discussion section (eg. 

Urbanowicz T et al. The Significance of Systemic Immune-Inflammatory Index for Mortality Prediction in Diabetic Patients Treated with Off-Pump Coronary Artery Bypass Surgery. Diagnostics (Basel). 2022 Mar 4;12(3):634. doi: 10.3390/diagnostics12030634. 

Zhang R, et al. Increased neutrophil count Is associated with the development of chronic kidney disease in patients with diabetes. J Diabetes. 2022 Jul;14(7):442-454. doi: 10.1111/1753-0407.13292. 

5.     The description of Table 2 should be changed because the authors presented different concentrations of markers in diabetic, DN and normal controls and did not evaluate prognostic value of these parameters 

6.     The manuscript lacks limitation and conclusion sections. Please add the relevant sentences.

7.     Language must be improved

8.     In some parts of the text two words are connected together eg “wasprepared”, ‘levelswere”

I am looking forward to get a possibility to read a corrected version. Best regards

Reviewer 

Author Response

Dear Reviewer 2 Thank you so much for the points to revised manuscript Kindly find the attached file containing answers of all the querries

Introduction part is according to manuscript all the recommended data incorporated except additional assays because current work was my students Ph.D work and she has already publish a paper containing fasting sugar, ldl, hdl, hb1Ac etc reference is included within the manuscript

rest of all the querries incorporated in the attached word fil

Round 2

Reviewer 1 Report

Dear Authors, 

A few comments have been revised based on the comments provided. 

However, the data on biochemical parameters are already been published in the Journal of Pharmaceutical Research International.

The data on biochemical parameters are the most important to be included for this research outcome. Authors, can not say, the report has already been published in another journal.

Moreover, tables and figures represented the same data. This is not revised. 

The subjects were categorized into three groups, T2DM with and without micro-albuminuria and controls. But, the authors did not mention the level of micro-albuminuria of the subjects involved. This is more important to discuss in the current manuscript. 

Authors can not incorporate the whole reference of number 10 in the text. 

Reviewer 2 Report

Thank you for partial implementation of the suggestions

The study results are included in study aim. It is not a common practice in the medical journals. 

Citation No 10 is inserted in the text

Still, the authors presented only relations between groups and not a prognostic value of the markers. Therefore, they are unable to conclude on the circulating stress markers as prognostic indicators determined in the study.